# Autophagy: A Potential Therapeutic Target to Tackle Drug Resistance in Multiple Myeloma

**DOI:** 10.3390/ijms24076019

**Published:** 2023-03-23

**Authors:** Hamed Bashiri, Hossein Tabatabaeian

**Affiliations:** 1Institute of Molecular and Cell Biology (IMCB), Agency of Science, Technology and Research (A*STAR), Singapore 138673, Singapore; 2Cancer Science Institute of Singapore, National University of Singapore, Singapore 117599, Singapore; 3Peter MacCallum Cancer Centre, Melbourne, VIC 3000, Australia

**Keywords:** autophagy, drug resistance, multiple myeloma

## Abstract

Multiple myeloma (MM) is the second most prevalent hematologic malignancy. In the past few years, the survival of MM patients has increased due to the emergence of novel drugs and combination therapies. Nevertheless, one of the significant obstacles in treating most MM patients is drug resistance, especially for individuals who have experienced relapses or developed resistance to such cutting-edge treatments. One of the critical processes in developing drug resistance in MM is autophagic activity, an intracellular self-digestive process. Several possible strategies of autophagy involvement in the induction of MM-drug resistance have been demonstrated thus far. In multiple myeloma, it has been shown that High mobility group box protein 1 (HMGB1)-dependent autophagy can contribute to drug resistance. Moreover, activation of autophagy via proteasome suppression induces drug resistance. Additionally, the effectiveness of clarithromycin as a supplemental drug in treating MM has been reported recently, in which autophagy blockage is proposed as one of the potential action mechanisms of CAM. Thus, a promising therapeutic approach that targets autophagy to trigger the death of MM cells and improve drug susceptibility could be considered. In this review, autophagy has been addressed as a survival strategy crucial for drug resistance in MM.

## 1. Introduction

Autophagy plays an essential role in the digestion and degradation of accumulated excess proteins and impaired or damaged organelles via a lysosome-dependent pathway [1]. Activation of autophagy during stress conditions such as starvation or production of reactive oxygen species (ROS) provides a source of energy and removes excess components, leading to cellular homeostasis and survival [2]. Nevertheless, during certain conditions, extensive autophagic activity may lead to type 2 programmed cell death (PCD) [3].

Three main types of autophagy that have been identified include microautophagy, chaperone-mediated autophagy (CMA), and macroautophagy, hereafter referred to as “autophagy”. The non-selective lysosomal degradative process that engulfs cellular components is known as microautophagy [4]. Direct membrane invagination forms vesicles [5] which, in turn, carry cell ingredients to the lysosomal vesicle, starting the breakdown of soluble cytoplasmic components or other completely integrated organelles, such as peroxisomes (Figure 1a). Proteostasis maintenance along with the cellular response to unfavorable situations are both facilitated by CMA [6]. In CMA, selective identification of substrates occurs in the absence of vesicles, followed by transfer through cytoplasmic hsc70/co-chaperones to the lysosomal membrane [7]. At the lysosomal surface, substrate internalization occurs by a membrane transfer compound that is created through multimerization of the CMA substrate–chaperone receptor, known as lysosome-associated membrane protein type 2A (LAMP-2A) [8].

The multimerization step leads to the translocation and degradation of the substrate (Figure 1b). During the normal state, very low levels of CMA exist in most cells [9], while stress factors such as starvation and oxidative stress upregulate CMA and enhance the number of lysosomes with active CMA that contain CMA substrates and hsc70 [10]. In research conducted by Nikesitch et al. [11], CMA was shown to increase in bortezomib-resistant MM, and its inhibition made bortezomib-resistant cells more susceptible. Moreover, there was a significant increase in the protein levels of LAMP2A, the rate-limiting component of the CMA pathway, in both MM patients who were resistant to bortezomib as well as in the bortezomib-resistant cell line model. Furthermore, compared to the parent cell line that was bortezomib-sensitive, the bortezomib-resistant cells had increased baseline CMA activities. Bortezomib-resistant cells became susceptible to bortezomib when CMA was inhibited, and the in vitro combination of bortezomib with CMA inhibition was more cytotoxic to myeloma cells than bortezomib alone. The findings of this study reveal that the elevation of CMA is a potential bortezomib-resistance mechanism and a new target for treating bortezomib-resistant MM. Autophagy is a highly conserved process regulated through various multistage signaling processes [12] to ensure the recycling of cytosolic organelles, proteins, macromolecules, and invasive microorganisms. During the autophagy process, an expanding bilayer structure called “autophagosome” isolates the substrates to be broken down in the lysosomes [13] (Figure 1c). Afterward, the degraded and digested components are transferred into the cytoplasm [14] to supply an alternate source of energy and maintain cellular homeostasis. During normal conditions, autophagy is activated to a small extent in most human cells to sustain homeostasis and/or to enhance cell viability in response to unfavorable situations [15]. Disruption of the regulatory mechanisms and/or mutations of autophagy genes triggers the occurrence and/or development of various human diseases and disorders, such as cancer. We will discuss the autophagy process as one of the tumor viability pathways in the context of multiple myeloma (MM), as it significantly affects the disease’s development and drug resistance.

## 2. The Autophagy Machinery and Its Regulatory Mechanisms

Several central markers interact during autophagic activity, among which are a group of genes known as AuTophaGy-related genes (*ATGs*). Thirty ATGs have been discovered so far in the yeast Saccharomyces cerevisiae [16], and their analogs have been identified in other eukaryotes. Constant expression of ATGs is required to regulate autophagy. The role and importance of Atg proteins in the formation of autophagosomes have been well investigated within different stages of autophagy [13,17]. As shown in Figure 1c, the autophagy process consists of multiple sequential steps, including initiation, elongation, maturation, fusion, and degradation. The mammalian target of rapamycin (mTOR) is considered the master regulator of autophagy [18,19] and, in eukaryotes, is a highly conserved protein kinase [20]. Upon interaction of mTOR with multiple proteins, two distinct complexes form—mTOR complex 1 (mTORC1) and 2 (mTORC2)—which have different reactivity to rapamycin, as well as different downstream outputs and upstream inputs [21,22]. Through the inhibition of catabolic activities (such as degradation of mRNA and activation of autophagy) on one hand, and by the activation of anabolic procedures (such as ribosome biogenesis, transcription, and protein synthesis) on the other hand, mTOR is involved in controlling cell growth [23]. Autophagy inducers, such as lack of oxygen, ROS, DNA damage, and starvation, negatively regulate the activation of mTOR [24] (Figure 2). In particular, mTORC1 is sensitive to starvation, which inactivates mTOR [25], leading to autophagy induction. Under stress situations, phosphorylation of Atg13 by mTOR does not occur, which results in the binding of Atg13 to Unc-51-like kinase-1/2 (ULK1/2) that then triggers the enzymatic capability to induce the formation of the phagophore [26,27]. Next, to complete the phagophore formation step, ULK1/2–Atg13 uses a kinase network [28] that includes UV radiation resistance-associated gene protein (UVRAG), vacuolar protein sorting-34 (Vps34), phosphatidylinositol-3-kinase class II (PI3KC3), Beclin-1, and autophagy and Beclin-1 regulator-1 (AMBRA-1) [29]. Beclin-1, a Bcl-2-homology (BH)-3 domain protein, is the mammalian orthologue of yeast Atg6 [30] and, depending on the type of the proteins (inducers or inhibitors) that bind to Beclin-1, functions as a central regulator to induce or inhibit autophagic activity [31] (Figure 2).

Phagophore elongation continues with two ubiquitin-like systems: the substrate receptor p62/sequestome1 (p62/SQSTM1) and the light chain3 (LC3) systems. Full-length LC3 can be found in the cytosol, and its activity depends on the processing of microtubule-associated proteins. Following autophagy stimulation, Atg4, a cysteine protease, proteolytically cleaves pro-LC3 to generate the soluble cytoplasmic form, LC3-I. Atg7, an E1-homologous factor, activates Atg4 with ATP to induce LC3-I movement to Atg3, an E2-homologous transfer factor [12]. Thereafter, active LC3-I conjugates with phosphatidylethanolamine (PE) to produce the lapidated form, LC3-II [32]. LC3-II can be detected on either the internal or external autophagosome membrane, and the interactions between Atg5–Atg12 determine its recruitment and combination. LC3-II is involved in the regulation of autophagosome–lysosome membrane fusion and the selection of cargo for digestion purposes [33]. p62/SQSTM1 is considered to be an autophagy marker and, through the LC3-connecting site, associates the protein ubiquitin-binding region to LC3-II to promote the recycling of ubiquitinated proteins at the autolysosome surface [34] (Figure 2).

Regulation of the autophagy process occurs in a variety of ways [35]. mTOR is the main regulator that functions like a signaling regulation site downstream of the insulin signaling, ATP content, and growth factor receptors. After starvation and reduced ATP levels, mTOR is suppressed and leads to the activation of adenosine 5′-monophosphate–activated protein kinase (AMPK) [36]. Moreover, inhibition of mTOR under the AMPK effect results in the activation of autophagy in a hypoxia-inducible factor (HIF)-independent and -dependent manner [37]. The endoplasmic reticulum (ER) stress response is triggered in the presence of hypoxia and through the unfolded protein response, which attenuates the mitochondrial mass along with the mitochondrial function in oxidative phosphorylation while enhancing autophagic activity to eliminate the ER-compacted portions [38]. This adaptation to hypoxia restrains the wasteful consumption of ATP by ER and limits the generation of ROS in the mitochondria. In addition, promoted autophagy can produce ATP from catabolism once there is limited ATP production by oxidative phosphorylation [39].

## 3. Significance of Autophagic Activity in the Survival of MM Cells

In tumor cells, metabolic stress triggers autophagy as an alternative source of energy and metabolites [40], promoting an adaptive cellular response to cancer treatments [41]. Autophagy is crucial in alleviating drug-induced cell death through chemoresistance in hematologic malignancies [42]. MM is a hematologic malignancy identified by the proliferation of monoclonal plasma cells in the bone marrow (BM). After non-Hodgkin’s lymphoma, MM is the second-most prevalent hematologic cancer (representing 10% of all hematologic cancers), substantially increasing in incident cases worldwide over the previous 25 years [43,44]. MM ontogeny is distinguished by various stages. The initial stage, defined as monoclonal gammopathy of unknown significance (MGUS), has no obvious signs, a low level of plasma cell (PC) replication, and low immunoglobulin production [45,46]. Individuals with MGUS have a 1% annual risk of developing MM by age 20 [47]. In some instances, it is feasible to diagnose a middle phase known as smoldering multiple myeloma (SMM), which has a larger immunoglobulin (Ig) production yet remains asymptomatic [48]. Once patients start to show symptoms, the disorder is known as MM, which may develop as an intramedullary or extramedullary condition [49,50]. The extramedullary condition is correlated with the worst outcome, particularly in the spreading phase of plasma cell leukemia, where high numbers of malignant PCs in peripheral blood circulation are detectable [51,52]. Monoclonal immunoglobulins (Igs), which are produced in large quantities by MM cells, cause potentially harmful misfolded or unfolded proteins to localize on the ER. Due to the increased proliferation capacity and Ig production, autophagy plays a critical role in the survival of MM cells to digest the extra protein aggregates [53]. Autophagy inhibition through the knock down of Beclin-1 expression or incubation in the presence of autophagy inhibitors, such as 3-methyladenine (3-MA) and chloroquine, results in the apoptosis of MM cells [54,55] and prevents autophagosome formation. Recently, Wang et al. [56] showed that elaiophylin, a potent inhibitor of the late autophagy phase, exhibits anti-MM cell activity through the inhibition of autophagic flux and sustained activation of ER stress-mediated apoptosis.

Moreover, basal autophagy is tightly controlled to avoid autophagic cell death. Lamy et al. [57] point to the heterodimeric protease caspase-10/FLIPL as a pro-survival factor that restricts basal autophagy via cleavage of the Bcl-2-interacting protein Bcl-2-associated transcription factor 1. Inhibition or silencing of caspase-10 stabilizes Bcl-2-associated transcription factor 1, which displaces Bcl-2 from Beclin-1, resulting in excessive autophagy and consequent MM cell death. Autophagy has been described in the literature as an MM pro-survival strategy that can provide a preventive impact during drug therapy, as drug-resistant MM cells are able to tolerate the cytotoxic effects of drugs through autophagy [58]. Bortezomib and carfilzomib are two proteasome inhibitors that are used as the primary agents for individuals with recently diagnosed or relapsed MM. Although these drugs exert an anti-cancer impact initially, patients frequently develop resistance. MM cells that resist bortezomib generate more autophagosomes and have more AMPK content than cells that are susceptible to the drug.

A decreased activity of AMPK impairs the generation of autophagosomes [59]. Riz et al. [60] showed that Kruppel-like factor 4, a transcription factor, binds to the *SQSTM1* gene promoter region and makes MM cells resistant to carfilzomib. According to Hoang et al. [61], exposing MM cells to inhibitors of mTOR and promoters of ER stress increases autophagy even more. mTOR is a famous autophagy inhibitor and treating MM cells with the drug inhibitor 3-MA results in a dose-dependent induction of autophagic cell death. Exposing MM cells to bortezomib along with autophagy inhibition simultaneously lead to synergistic toxic effects [61]. Suppression of the PI3K/Akt/mTOR signaling pathway positively correlates with the activation of autophagy under ER stress [62]. Fu et al. [63] showed that ER stress enhances autophagic activity and apoptosis while decreasing cell expansion via suppression of the PI3K/Akt/mTOR pathway in a large cohort of MM individuals classified as susceptible and resistant patients depending on the effectiveness of the chemotherapeutic approach. In addition, ER stress may return DR through the PI3K/Akt/mTOR axis. Inside the cells, nicotinamide adenine nucleotide is critical in regulating various cellular processes. It is highly expressed in MM cells and is involved in drug resistance as well as cell growth [64]. Cea et al. showed that suppression of nicotinamide phosphoribosyltransferase (NAMPT), a rate-limiting enzyme engaged in nicotinamide adenine nucleotide production, evoked related cytotoxic effects against MM cells resistant to routine anti-MM drugs in vitro and in vivo, and inhibits the preventive properties of IGF-1, interleukin-6, and BM stromal cells. The cytotoxicity of the NAMPT inhibitor FK866 is caused by autophagy activation via inhibition of the mTORC1/Akt and ERK1/2 pathways [64]. In cancers, autophagy plays dual roles. Numerous cancers, including pancreatic, hepatic, and colorectal cancers, along with hematologic malignancies such as lymphoma, leukemia, and myeloma have been linked to impaired autophagic activity.

Additionally, changes in autophagic activity can lead to the development of drug resistance upon undergoing chemotherapy with drugs such as doxorubicin, etoposide and cisplatin [65]. Nevertheless, as autophagy has both tumorigenic and tumor-suppressive impacts, its significance in cancer is not precisely known [4,40]. In tumors, autophagy-related genes are frequently absent. For example, *Beclin1*, a gene that produces a crucial protein member of the PI3K complex, is typically downregulated in human breast cancer and carries monoallelic deletions [66]. Moreover, it has been proposed that Beclin1 and its positive activator, UV radiation resistance-associated gene (UVRAG), are crucial in activating autophagy and inhibiting tumor formation and proliferation [67]. Autophagy is involved in oncogenesis as it affects the adaptation of tumoral cells to stress conditions, such as ischemia, where it is localized in the tumor center to support malignant cells with the essential nutrients for their expansion prior to the initiation of angiogenesis [68]. Moreover, the loss of the tumor repressor enzyme phosphatase and tensin analog (PTEN) activates Akt, which results in a significant reduction in the autophagy of malignant cells [69].

Notably, in MM, autophagy might serve as a pro-survival strategy that helps tumor cells to remove the enormous accumulation of harmful, misfolded Igs. Additionally, autophagy helps myeloma cells resist proteasome antagonists. Proteasome inhibition causes ER overloading and stress by increasing the aggregation of damaged proteins in the intracellular milieu. Proteasome inhibition also promotes autophagic activity, resulting in drug resistance, which is consistent with the close relationship between cell stress, autophagy, and apoptosis. Hence, a novel treatment approach that targets autophagy to trigger the death of myeloma cells and enhance drug susceptibility may be considered. Particularly, targeting autophagy may concentrate on activating or inhibiting autophagic activity due to its dual function as a process of pro-survival or cell death [70]. Moreover, various stress factors in the tumor microenvironment, such as hypoxia, starvation, inflammation, and extracellular matrix breakdown, activate pro-survival autophagy.

The evolution of chemoresistance in malignancies is a significant issue in the clinic. The development of autophagy inhibitors, intended to boost chemosensitivity, has been hugely influenced by this [71]. Phase I/II clinical trials using hydroxychloroquine (HCQ)-mediated autophagy repression have been performed recently in a number of malignancies, such as myeloma, pancreatic cancer, and melanoma [72,73,74,75]. These studies assessed patient toxicity, clinical activity, maximum tolerated dosages, and pharmacodynamics. It has been demonstrated that the antimalarial medicine HCQ inhibits the end stage of autophagic activity [76]. In a phase I clinical trial, the mTOR inhibitor temsirolimus and HCQ were used to treat patients with melanoma and advanced solid tumors. Although there was no response, the majority of patients who received treatment had stable illnesses. The median progression-free survival in 13 patients with melanoma who received HCQ at 1200 mg/d in addition to temsirolimus was 3.5 months [73]. Moreover, pancreatic adenocarcinoma was the subject of HCQ clinical studies during which HCQ had a minor anti-tumor effect due to the unstable inhibition of autophagy [74]. Due to its efficiency, the proteasome inhibitor bortezomib has become the standard of care treatment for multiple myeloma [77]. In 22 individuals with refractory or relapsed myeloma, bortezomib and HCQ produced 14% partial responses, 14% modest responses, and 45% persistent illness [75]. Consistent with other studies, the HCQ with bortezomib combination led to an elevation of autophagosomes as a pharmacodynamic indicator of autophagic activity modification. The abovementioned clinical trials demonstrate that it is possible to achieve therapy-induced autophagy suppression and that significant advancements have been made in the manipulation of autophagy for cancer treatment.

Frassanito et al. [78] proposed a new strategy that moderates the interaction between MM cells and cancer-associated fibroblasts (CAFs) in MM drug resistance. CAFs are critical cells inside the BM microenvironment and enhance cancer formation, development, and drug resistance [79]. MM CAFs co-cultured with MM cells shows resistance to bortezomib in vitro, indicating that MM CAFs inhibit bortezomib-induced apoptosis. The authors have demonstrated that bortezomib treatment activates autophagy in MM CAFs by inhibiting mTOR, inducing LC3- II, and activating the transforming growth factor beta (TGF-β) pathway [64]. Autophagy inhibition by knockdown of Atg7 using small-interfering RNA or treatment with 3 MA or TGF-β inhibitor restores sensitivity to bortezomib in bortezomib-resistant CAFs and leads to cytotoxicity in MM cells co-cultured with CAFs.

## 4. Multiple Myeloma Drug Resistance and Autophagy

Various stress conditions, including hypoxia, starvation, extracellular matrix reduction, and inflammation in the tumor tissue, activate survival-promoting autophagy. The emergence of proteasome inhibitors (PIs) and immunomodulatory drugs has significantly developed the prognosis of MM patients. Bortezomib prevents MM cell proliferation, leads to apoptosis, and disturbs MM cell crosstalk with the BM stroma by inhibiting cytokine circuits [80,81]. Even though the efficacy of bortezomib is proven in MM patients, relapse caused by bortezomib resistance is unavoidable, and the malignancy is still untreatable [82]. Bortezomib resistance is thought to be caused by the induction of autophagy, as bortezomib increases the aggregation of polyubiquitinated proteins [63,82]. Protein aggregation leads to the formation of aggresomes and autophagosomes, which may increase protein degradation, tumor viability, and relative resistance to drugs (Figure 3). Bortezomib interacts with cancer-associated fibroblasts (CAFs) in the BM microenvironment to trigger ROS and autophagy by blocking mTOR and p62 [63,82]. CAFs are critical in the BM microenvironment and enhance cancer formation, development, and drug resistance. In vitro co-cultures of MM CAFs and MM cells resist bortezomib, indicating that MM CAFs inhibit the apoptosis caused by bortezomib [83].

Moreover, bortezomib treatment has been shown to promote autophagy in myeloma CAFs by inhibiting p62 and mTOR, inducing LC3, and activating TGF-b [63] (Figure 3). Enhanced autophagy prevents the harmful effects of proteasome inhibition and inhibits apoptosis. This potential drug resistance is an essential obstacle to therapy and the survival of individuals with MM. As discussed earlier, no treatment for MM is known so far. As resistance to bortezomib evolves, scientists are seeking approaches to conquer this problem by understanding major resistance strategies as potential therapeutic targets [84]. Regarding the pathophysiology of resistance to bortezomib, autophagy is crucial in preventing apoptosis, despite enhanced stress [82]. As mentioned earlier, caspase 10 is an autophagy regulator and inhibits cell death induced by autophagy [85]. Once the mechanism for attenuating the pro-survival effects of autophagy is compromised, the autophagy triggered by bortezomib treatment and resistance might result in cell death. Further exploration into improving and utilizing caspase-10 suppressors such as Z-DEVD-FMK might be advantageous in combination with bortezomib to drive MM-resistant cells into death [86].

Since bortezomib was approved in 2003 to be used in the therapies of resistant/relapsed multiple myeloma (MM) [87], and with the subsequent agreement to apply bortezomib as the first-line treatment of MM in 2008, the use of PIs for therapeutic purposes in blood cancers has increased dramatically [88]. In comparison with standard chemotherapy strategies, patients with MM malignancy lived twice as long after the development of autologous stem cell transplantation [89]. Bortezomib was authorized in 2006 to treat individuals with relapsed mantle cell lymphoma (MCL) [90]. Later, new PIs were produced to improve bortezomib’s oral bioavailability, lessen unfavorable effects, and address resistance to bortezomib occurrence [91]. Then, the new PI carfilzomib was subsequently authorized to treat MM individuals who relapsed after receiving bortezomib plus an immunomodulatory medication (IMiD) [92,93]. Encouraging outcomes of PIs in individuals with MM and MCL pave the way for testing their action in more hard-to-treat hematologic cancers having low survivability, such as acute leukemia [94]. In spite of the positive outcomes of bortezomib therapy, the emergence of the bortezomib resistance phenomenon is a growing barrier that hampers the drug’s treatment efficacy [95,96,97]. Therefore, understanding and managing the complex processes behind both innate and adaptive resistance to PIs is essential to improve their therapeutic effectiveness. Alongside proteasomal activity, autophagy provides an alternate approach to recycling and breaking down proteins inside the cells. Autophagy and ubiquitin–proteasome system (UPS) were formerly believed to act separately, but it has recently been discovered that these two proteolytic mechanisms work together. In order to degrade cytosolic proteins, autophagy acts through the fusion of lysosomes with double-layer structures called autophagosomes. Numerous human diseases, including cancer and neurological disorders, are related to disturbed autophagy [98]. It seems that autophagy functions as a tumor suppressor in the normal state, while it can promote tumor cell viability during stress situations [99]. It appears that by inhibiting the proteasome, autophagy is activated as a survival strategy for the removal of UPS substrates [100]; therefore, activated autophagy may be involved in the resistance to bortezomib [101]. Accordingly, activating transcription factor 4 (ATF4), which induces autophagic activity, was elevated after the proteasome was inhibited [102,103,104]. Hence, it is possible that simultaneous suppression of autophagy and the proteasome could synergistically lead to cell death. A combination of autophagy and proteasome suppressors has increased cell death compared to single-treatment strategies [104,105].

Moreover, Jia et al. demonstrated that induction of autophagic activity upon bortezomib treatment in diffuse B-cell lymphoma (DLBCL) cells resulted in moderate resistance to bortezomib. In contrast, bortezomib treatment and the lysosomotropic factor chloroquine, a protein denaturing inhibitor in the phagolysosome, synergistically affect cell death [106]. Clinical trials have been performed to evaluate the effects of chloroquine on solid tumors. In addition, in the phase II clinical trial (NCT01438177), the combined effects of cyclophosphamide, bortezomib, and chloroquine have been examined on individuals with refractory myeloma who had progression upon combined bortezomib/cyclophosphamide treatment. Eight participants received at least two treatment cycles; 3/8 had a partial response, 1/8 had stable disease, and 4/8 had worsened, with a 40% clinical benefit rate. According to the present studies’ findings, a combination of chloroquine, cyclophosphamide and bortezomib, helps overcome resistance to proteasome inhibitors in a considerable number of patients who have received an intensive pre-treatment regimen while minimizing side effects [107]. In this regard, more special drug autophagy suppressors should be examined. To recreate the patient’s tumor growth and to test alternative therapies more accurately, further efficient treatments could also involve 3D models, depending on the individual’s BM microenvironment [86,108]. In addition, the 3D models may be applied to strictly follow-up disease development so that therapeutic strategy can be quickly amended [108].

## 5. Effects of HDAC Inhibitors on the Promotion of Autophagic Activity

Histone acetylation is a major controlling process and is involved in the transcription regulation of nearly 2–10% of genes [109]. Although enhanced acetylation leads to chromatin decondensation, histone deacetylation results in the condensation of chromatin [110]. These modifications may lead to reduced or enhanced transcription of genes. Nevertheless, besides histones, more proteins exist, such as transcription coregulators, structural proteins, and mediators of signaling, whose function is under the influence of acetylation. Specifically, the activity of transcription factors may be enhanced or reduced, which explains why gene expression affected by histone deacetylases (HDAC) suppression is not upregulated at all times, even in loosened chromatin structures. Since Histone deacetylase 6 (HDAC6) provides a mechanical link between autophagy and UPS through the transfer of accumulated proteins to aggresomes, it can be considered a promising target for treatment [111]. Bortezomib has been shown to increase the formation of aggresomes, which might play a role in the transportation of (ubiquitinated) accumulated proteins to lysosomes through autophagy [112]. Lack of HDAC6 has been recognized to be associated with the failure to remove accumulated proteins and the formation of large aggresomes. Treatment with bortezomib combined with HDAC suppressors had synergistic effects in preclinical models [113] and clinical trials in MM individuals [114]. Moreover, therapeutic strategies combining new PIs and HDAC suppressors are under evaluation in the preclinical stage [115,116,117]. The ratio of histone acetyltransferases (HAT) and HDACs determine the acetylation of most autophagy-related proteins, including the proteins produced by ATGs [118]. Acetylation of the Forkhead Box O1 (FOXO) family of transcription factors is another regulatory mechanism of autophagic activity [119]. Various HDACs regulate autophagy through various approaches, as evidenced by the findings of a number of studies detailed herein. HDAC6 promotes autophagic activity once there is an impairment in the ubiquitin–proteasome system (UPS). HDAC2 knockdown in cardiomyocytes prevents autophagy [120]., while HDAC1 knockdown in HeLa cells increases autophagosome formation [121]. HDAC10 enhances cell viability of neuroblastoma cells (IMR32, Kelly, and E(2)-C cells), and its suppression makes the cells sensitive to cytostatic drugs [122]. Sirtuin 1 (SIRT1) triggers autophagic activity by forming a network with Atg5, -7, and -8, members of the autophagic process [123]. There is still controversy over the function of HDAC suppressor-induced autophagy in tumor cell death. Autophagy has been shown in some research to contribute to cell death, as inhibiting autophagy or depletion of ATGs decreases the effectiveness of HDAC suppressors, which are used as anti-cancer drugs. Studies using in vivo models have found that combining HDAC suppression with autophagy blockade can reduce the growth of colon cancer cells (HCT-116) [124]. On the contrary, autophagy-mediated digestion of substrates inside the cells is considered to be a cell death signal and can result in the toxic effects of autophagy. As an example, in a group of hepatocellular carcinoma cells, the cytotoxicity induced by SAHA (an HDAC inhibitor) can be prevented by 3-methyladenine, which blocks autophagy by inhibiting the formation of autophagosomes via class III PI3K inhibition, or by knocking out the Atg5 gene [125]. Cell death of endometrial stromal sarcoma cells treated with SAHA has been attributed to autophagy [126]. Cancer cells with wild type TP53 undergo apoptosis when exposed to SAHA. However, when TP53 is absent or degraded, the autophagic pathway is activated, which results in cell death [127]. The differences, as mentioned earlier, may arise from variations in the tumor cells, cancer models, HDAC suppressors, and HDAC suppressor dosages. Multiple signaling pathways are involved in activating autophagy through the inhibition of HDAC. A main autophagy inhibitor is mTOR (mammalian target of rapamycin) which acts by phosphorylating and inactivating the Unc-51-like autophagy activating kinase 1ULK1 that is considered to be an upstream target within the autophagy process. Therefore, ULK1 activities are reversed by SAHA’s deactivation of mTOR [125,128,129]. The increase in *ATG* gene production brought on by SAHA is a result of the activation of NF-kB through the adjustment of the RelA/p65 (a component of the NF-kB) signaling pathway [130]. Researchers have found that SAHA induces autophagy in cells from leukemic and hepatocellular carcinoma patients through ROS production [129,131]. One crucial aspect for clinical use appears to be the ability of some HDAC suppressors to promote autophagic activity, which leads to cell death in cells resistant to apoptosis. In HeLa cells, romidepsin and HDAC1 siRNA enhance autophagic activity [121]. SAHA induces autophagic activity by downregulating the AKT–mTOR axis, which results in growth inhibition of short-term culture glioblastoma cell xenografts in nude mice [132]. According to the aforementioned findings, activation of autophagy via HDAC suppressors might be a potential anti-cancer therapeutic approach.

## 6. Effect of CAM, an Autophagy Antagonist, on MM

Clarithromycin (CAM) is a semisynthetic macrolide antibiotic and a popular bactericidal medication [133]. Recent studies have recognized the effectiveness of CAM as a supplemental medication for the treatment of MM. Even though the treatment of MM with CAM alone has not been successful, combined chemotherapies that included CAM have proven effective [134]. It is well recognized that a number of myeloma growth factors (MGFs), notably interleukin (IL)-6, play a crucial role in the progression of MM. In addition, it has been demonstrated that CAM inhibits various MGFs, most notably IL-6. The inhibition of MGFs, autophagy suppression, reversion of drug resistance, immunoregulatory impact, and steroid-sparing/enhancing effect have all been proposed as potential action mechanisms of CAM in treating MM [135,136,137]. Moreover, the hallmark of MM is the unregulated proliferation of plasma cells that produce monoclonal immunoglobulin (Ig). Therefore, the generation of high amounts of misfolded or unfolded Ig may result in significant stress on the ER [138,139,140]. Hence, MM begins as a delicate malignancy that is especially vulnerable to autophagy-, proteasome-, and histone deacetylase-6 inhibitors. Together, CAM’s synergistic effects contribute significantly to the therapy of MM.

It is widely recognized that CAM inhibits autophagic activity effectively and continuously. In vitro analysis of the influence of CAM on MM cells was performed by Nakamura et al. [141]. At clinically relevant doses (6–50 g/mL), they showed that CAM inhibited the autophagic process by preventing the late stage of the autophagic activity, likely following the autophagosome–lysosome fusion step i.e., CAM inhibits MM cell proliferation by stopping the autophagy process. Therefore, CAM may be employed as an adjuvant in MM therapy regimens when the tumor uses autophagy to escape apoptosis.

One of the hallmarks of MM is the hyperproliferation of malignant monoclonal plasma cells, which produce large amounts of immunoglobulins and increase the accumulation of misfolded or unfolded proteins. The accumulation of misfolded or unfolded proteins leads to ER stress and the unfolded protein response (UPR) [138,139,140]. Based on the Obeng et al. findings [140], MM cells are intrinsically vulnerable to proteasome antagonists as they produce a lot of Igs, which necessitates the expression of physiologic *UPR* genes. UPR stimulation causes cell-cyclic inhibition and apoptosis induction in MM cells if the ER stress is solid or persistent.

## 7. Effects of CAM in Combination with a Proteasome Suppressor or an Antagonist of HDAC6

Bortezomib, the first agent in the class of proteasome antagonists of the 26S proteasome, was formerly accepted to be used as a monotherapy drug in the therapy of individuals with relapsed or refractory multiple myeloma (RRMM) [142]. The mechanism of action of bortezomib has been found to be a consequence of its inhibitory effect on the ubiquitin–proteasome process, which results in the accumulation of misfolded or unfolded proteins in myeloma cells. This triggers ER stress as well as the UPR [139]. The UPR triggers the chaperone protein GRP-78 to be activated in order to preserve ER function. Moreover, it increases the expression of the transcription factor CHOP (i.e., C/EBP homologous protein), allowing cell death when ER stress is excessive for the cell’s adaptability to tolerate [140]. The combination of bortezomib’s suppression of the ubiquitin–proteasome system and CAM’s suppression of the autophagy–lysosome system have a synergistic effect on the stimulation of the UPR, which leads to MM cell death (Figure 4). According to an experimental study by Moriya et al., using CAM and bortezomib together led to more cytotoxic effects than monotherapy with Bortezomib [93]. Recently, clinical studies showed the efficacy of combination therapy, including bortezomib, CAM, and lenalidomide [143]. Once vorinostat (HDAC6 antagonist), which is recognized to prevent aggresome synthesis, was combined with bortezomib and CAM, the apoptosis-inducing impact was improved further [139] (Figure 4). Nevertheless, clinical trials utilizing this combination have not been conducted thus far.

## 8. HMGB1-Induced Chemoresistance in MM

High mobility group box 1 (HMGB1) is a nonhistone chromatin-associated protein that has been widely reported to play a pivotal role in the pathogenesis of hematopoietic malignancies [144]. MM patients with elevated levels of HMGB1 have a lower 3-year survival rate, which could be linked to increased resistance to MM treatment. HMGB1 is involved in the autophagy and DNA damage repair processes. On the other hand, once HMGB1 is decreased, the mTOR pathway is activated to suppress autophagy and trigger apoptosis, increasing the susceptibility of myeloma cells to dexamethasone (Dex) [145]. Likewise, Gao et al. discovered that metastasis associated lung adenocarcinoma transcript 1 (MALAT-1) expression and HMGB1 protein levels were drastically reduced in individuals in complete recovery compared with those with untreated MM, where both lncRNA MALAT-1 and HMGB1 levels were significantly elevated. Additionally, MALAT-1 induces HMGB1 ubiquitination in MM cells to increase HMGB1 expression post-translationally, which promotes autophagic activity and prevents apoptosis [146].

Moreover, Roy et al. found that HMGB1 levels were elevated in bortezomib-resistant myeloma cells and that bortezomib and lycorine effectively resensitized refractory cells to bortezomib. The MEK–ERK pathway is inactivated through proteasomal degradation of HMGB1 by lycorine, preventing Bcl-2 from dissociating from Beclin-1 and subsequently suppressing autophagy [147]. Hence, HMGB1 is a crucial target for individuals with MM to improve drug sensitivity during chemotherapy. A serious impediment to the therapeutic strategies of hematological malignancies is acquired chemoresistance. Numerous studies have shown that chemotherapy drugs such as methotrexate, cisplatin, etoposide, docetaxel, and doxorubicin (DNR) stimulate HMGB1 overexpression and enhance HMGB1 translocation to cytosol [148,149,150]. In MM, decreased levels of HMGB1 increase bortezomib activity while inhibiting autophagy [147]. In times of altering nutrition supply, autophagy is a degrading system that modifies cells to maintain their energy level [151].

Nevertheless, over time and during stressful situations, including starvation and lack of oxygen, autophagy increases the viability of tumor cells [152]. Therefore, inhibiting autophagy will make cancer cells more susceptible to chemotherapy. Three mechanisms exist by which HMGB1-dependent autophagy enhances chemotherapeutic resistance: (1) nuclear HMGB1 increases the heat shock protein 27 (HSP27) expression level; (2) extracellular HMGB1 attaches to receptor for advanced glycation endproducts (RAGE); and (3) the Beclin-1/PI3K-III compound is activated by HMGB1 [153,154]. Hematopoietic malignancies could be treated by targeting HMGB1, which would inhibit autophagy [155,156]. The separation and recoupling of autophagic components are crucial steps in autophagy-related chemoresistance. In leukemic cells, *HMGB1* gene transfection can raise LC3-II contents and suppress the mTORC1 pathway to trigger autophagic activity and induce chemoresistance [157,158]. In tumor cells undergoing death, HMGB1 is secreted, and RAGE-mediated ERK/Drp1 phosphorylation improves autophagy-induced chemoresistance and regeneration. By inhibiting autophagy, HMGB1 and RAGE antagonists eliminate Drp1 phosphorylation and dramatically increase susceptibility to chemotherapy [159]. Additionally, employing chemotherapy agents on leukemic cells causes the translocation of HMGB1, which is engaged in autophagy and finally enhances chemoresistance in hematologic malignancy [160,161].

## 9. Rescue from Autophagic Activity, Disrupted Intracellular Signaling Processes, and Apoptosis

Resistance against drug-induced apoptosis is another common strategy of treatment resistance identified in individuals with MM. Key signaling pathways, including nuclear factor kappa B (NFkB), PI3K/AKT, and the proteasome pathway, all play a vital role in the process of programmed death of cells, known as apoptosis [162]. Apoptosis in MM cells is primarily induced through the IL-6-activated Janus kinase/signal transducer and activator of transcription 3 (JAK/STAT3) and MAPK/ERK pathways [163]. Other mediators, including VEGF, insulin-like growth factor 1 (IGF-1), stromal cell-derived factor 1 (SDF-1), and fibroblast growth factor (FGF), can similarly activate the MAPK/ERK and PI3/AKT pathways [164,165]. Human cells and myeloma cell lines resistant to doxorubicin, mitoxantrone, dexamethasone, and melphalan undergo apoptosis by an Apoptosis 2 ligand/tumor necrosis factor-related apoptosis-inducing ligand (Apo2L/TRAIL) pathway [166].

Additionally, via enhancing apoptosis, Apo2L/TRAIL activation reversed resistance to bortezomib in MM cells [167]. Mcl-1, a protein that promotes survival, is involved in the survival of myeloma cells. Its suppression quickly caused apoptosis in MM cells, while its overexpression increased the risk of recurrence and aggravated the malignancy at different phases [168,169,170]. Upon stimulation of the JAK/STAT3 pathway, Mcl-1 expression was observed to be increased in myeloma cell lines [171], and in primary cells and cell lines once VEGF was available [172,173].

Some of the MM cell lines as well as primary cells showed an association between the MM phenotype and elevated Bcl-2 expression along with reduced Bax levels [174]. Elevated concentrations of the Bcl-XL protein inhibited apoptosis by activating the JAK/STAT3 pathway through IL-6. Bcl-XL levels appear to be linked to MM drug resistance, as higher amounts were observed in those patients who had relapsed in comparison with those who had just been diagnosed [175].

NFkB, which consists of a group of five transcription factors, is well recognized for preventing apoptosis, which helps tumor cells to survive. NFkB is critical in the pathogenesis of MM and has been discovered to be continuously functional in clinical specimens and myeloma cell lines [176,177]. Additionally, it has been shown that NFkB contents were greater in MM cells taken from individuals in the relapsed phase, and also that drug-susceptible MM cells exhibit lesser NFkB activity in comparison to those which are resistant to drugs [178]. Therefore, IkB kinase inhibitors, bortezomib, or arsenic trioxide have been used in a variety of investigations to examine the effects of NFkB blockade on myeloma cell lines by promoting apoptosis [177].

As high levels of unfolded/misfolded proteins exist in MM cells, they very much depend upon the unfolded protein response (UPR) mechanism to return to homeostatic conditions. In order to reduce ER stress, the UPR is recruited, which reduces the synthesis of proteins and promotes the transcription of chaperones that are involved in the folding of heat shock proteins (HSPs) [179]. Then, the residual misfolded proteins inside the ER will be degraded through autophagy and proteasome processes [140,180,181]. Various transcription factors reach the nucleus and stimulate UPR target genes to control the UPR pathway [181]. This dependency of MM cells on the UPR pathway and its genes makes them more susceptible to PIs. As an example, bortezomib has a powerful effect on MM cells as it prevents proteasome function, leading to the aggregation of misfolded proteins in the ER, which is lethal for tumor cells and causes them to undergo apoptosis [140]. However, some individuals become resistant to bortezomib. There is evidence that levels of the UPR central transcription factor, X-box binding protein (XBP1), and bortezomib responsiveness are correlated.

Increased XBP1 levels were associated with greater susceptibility to bortezomib [182]. Additionally, in vitro research revealed that lower activating transcription factor 6 (ATF6) expression, a UPR regulator and a stimulator of the XBP1, as well as smaller ER sizes, are associated with resistance to bortezomib. Altogether, these findings point to a potential relationship between reduced UPR function and resistance to bortezomib; however, more research is required to confirm this relationship in a clinical setting [179].

Autophagy induction is crucial in MM cells as it contributes to the UPR response and enables the cells to survive by allowing them to destroy misfolded proteins. Therefore, autophagic activity is accompanied by resistance to drugs in MM. Once the autophagy-stimulator activating transcription factor 4 (ATF4) was shown to be elevated after treating several cancer cell lines with a proteasome antagonist, it was discovered that autophagy had a role in resistance to bortezomib [102]. Accordingly, approaches to inhibit autophagy have been investigated. Some strategies aim to block autophagy to trigger apoptosis after medication administration. Autophagy blockers and bortezomib together demonstrated positive outcomes in phase I and phase II clinical studies for the therapy of patients who had relapsed or were drug-resistant [75,107]. Moreover, in vitro and in vivo studies have shown that autophagy antagonists and carfilzomib synergistically increase apoptosis induction [183,184].

## 10. Conclusions

Drug resistance has remained a main hindrance to improving cancer patients’ survival [185]. Autophagy has emerged as a critical mechanism in developing drug resistance in multiple myeloma, a malignancy of plasma cells. Autophagy is a cellular process by which cells break down and recycle cellular components, including damaged proteins and organelles. In multiple myeloma, autophagy has been shown to promote cell survival and drug resistance by removing the toxic effects of chemotherapy drugs, thereby allowing cancer cells to persist and continue to grow despite treatment. Therefore, targeting autophagy should be considered a promising strategy for overcoming drug resistance and improving the effectiveness of treatment for multiple myeloma. However, further research is needed to fully understand the role of autophagy in MM and to develop targeted therapies that can effectively inhibit this process.

## Figures and Tables

**Figure 1 ijms-24-06019-f001:**
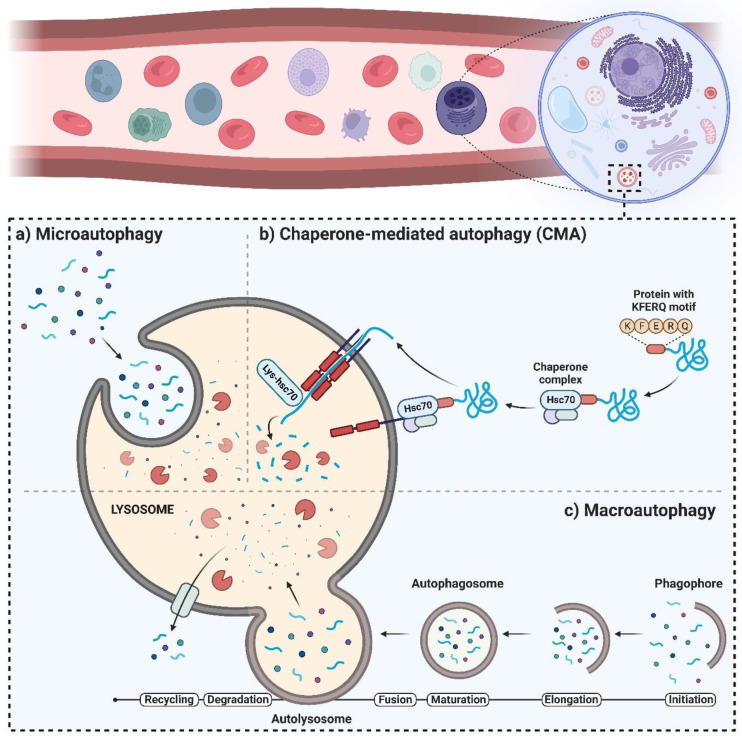
Different types of autophagy: (**a**) microautophagy, (**b**) chaperone-mediated autophagy (CMA), and (**c**) macroautophagy.

**Figure 2 ijms-24-06019-f002:**
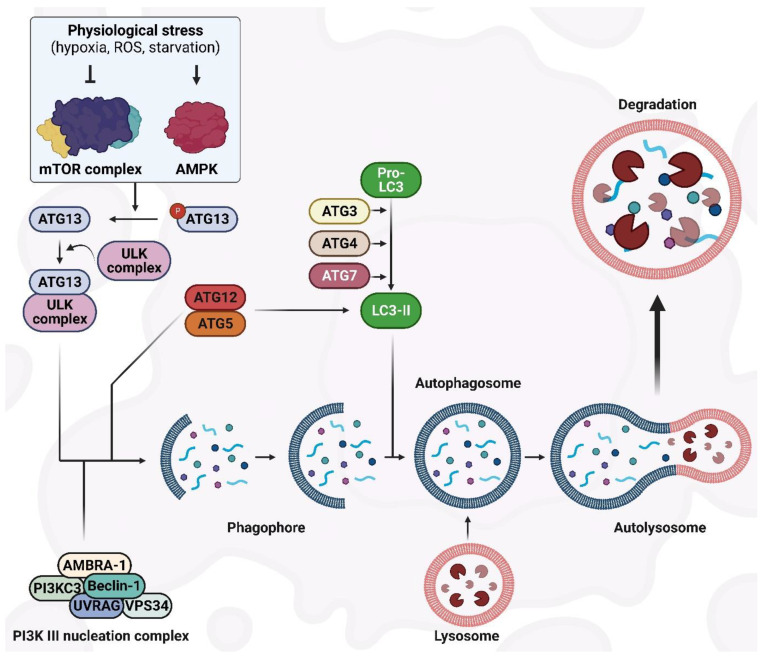
The autophagy machinery and its regulatory mechanisms.

**Figure 3 ijms-24-06019-f003:**
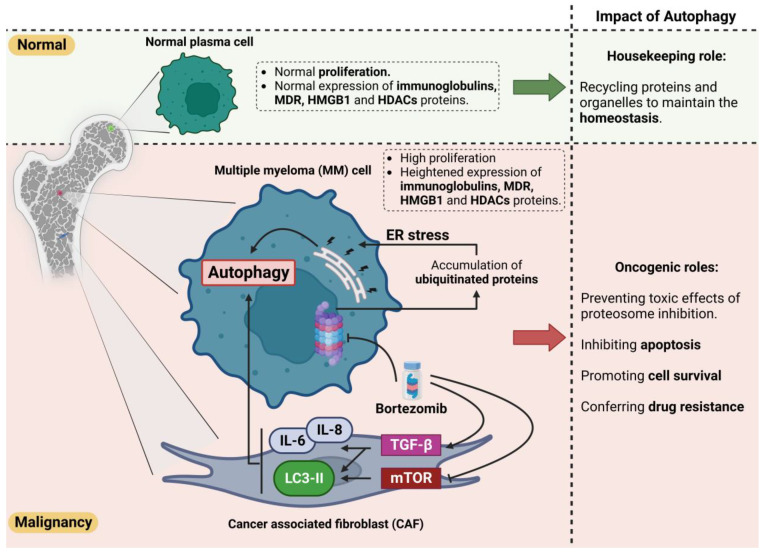
Dual role of autophagy in normal and malignancy conditions. Autophagy plays a protective role in the homeostasis of normal cells via recycling of the aggregated/unfolded/misfolded proteins and defective organelles. However, this protective effect could enhance the survival of tumor cells by clearing the accumulated proteins and improving their turnover, thereby inhibiting apoptosis. Moreover, autophagy could cause resistance to bortezomib by inducing ER stress in MM cells and triggering the pro-autophagic pathways in cancer-associated fibroblast cells (CAFs). Heightened autophagy clears the bortezomib-induced accumulation of ubiquitinated proteins. The MM cells could subsequently escape from apoptotic pathways and survive during treatment.

**Figure 4 ijms-24-06019-f004:**
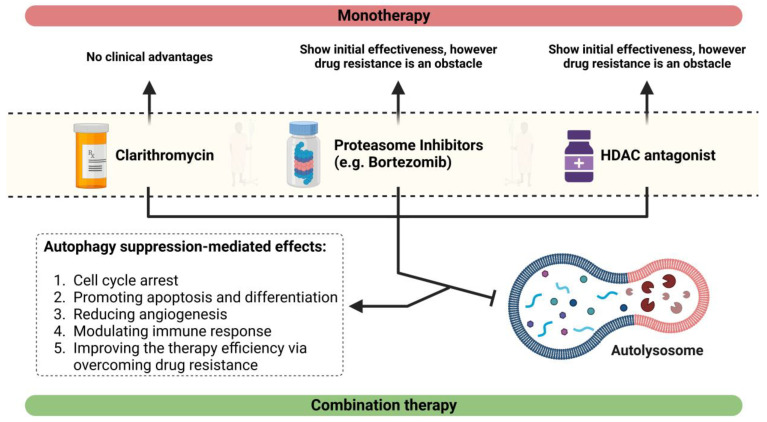
Inhibition of autophagy via combination therapy could improve the treatment efficiency of MM patients. Upon suppression of autophagy, proteasome inhibitors such as bortezomib could induce apoptosis and subsequently rein in cancer development.

## Data Availability

Not applicable.

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
