# Peer review of "Autophagy: A Potential Therapeutic Target to Tackle Drug Resistance in Multiple Myeloma"

_ijms, 2023, doi:10.3390/ijms24076019_

Round 1

Reviewer 1 Report

In review by Bashiri and Tabatabaeian the authors concentrate on autophagy as a major factor in multiple myeloma drug resistance (MM). Recent studies have shown that autophagy plays a role in the emergence of MM drug resistance, including HMGB1-dependent autophagy, proteasome inhibition, and the use of clarithromycin as adjuvant. Targeting autophagy may be a potential therapeutic strategy to cause the death of MM cells and increase drug susceptibility. Therefore, this review could be suitable for publication in IJMC, but after some improvements:

1.    In section 3 the authors need to describe some details about multiple myeloma (MM). The process of autophagy is explained in so many details, but multiple myeloma only in one sentence. This is review about autophagy involvement in MM drug resistance and authors have enough space to introduce readers with some basics about MM.

2.      Moreover, in second paragraph of this section (references 52 and 53) it seems that autophagy activation do not have prosurvival role, but in contrast decrease MM cell expansion or increase cytotoxicity of inhibitor FK866. Therefore, in this section authors should emphasize dual role of autophagy on survival of MM.

3.      In this review there are several structural problems that make review little bit confusing. Section 4 is dedicated to MM drug resistance and autophagy but here is only one example about bortezomib resistance. Since section 5 is strictly about the role of autophagy in bortezomib resistance my suggestion is that authors fuse these two sections since they are practically the same. Next, at the end of section 5 authors introduced HDAC, their inhibitors and combination with bortezomib. This text should be moved to section 6 which is about HDAC inhibitors especially since this text is only example about HDAC inhibitors in the treatment of MM. Also describe in the text what is SAHA, is it HDAC inhibitor?

4.      I suggest that authors write more precisely about NCT01438177 clinical trial (mentioned in section 5) since we do not know whether this study is active, terminated, are there promising results etc.

Reviewer 2 Report

This review article addresses the function of autophagy in drug resistance and describes the detailed mechanism of the resistance pathway in myeloma. However, the article still had some concerns that need to be addressed as below:

1.     In sections 2 and 3, the authors need to draw a figure to address the overall mechanism of resistance and autophagy, especially in myeloma. In these two sections, it is hard to follow the description.

2.     In sections 7, 8 and 9, the authors described the combination therapy to induce cancer cell death through the autophagy pathway. In this part, the authors could conclude by the schematic diagram. It will help the manuscript more clearly to understand.

The authors mentioned proteasome inhibitor is critical in autophagy regulation. However, no figures show the mechanisms of these two-pathway relationships. The authors may draw a figure to show a clear mechanism in myeloma under bortezomib treatment.

3.     The authors mentioned microautophagy and macroautophagy in the introduction. However, how they regulate resistance in myeloma still needs to be included. Did any previous study show the results of these two autophagy mechanisms in myeloma?

4.     It already had relative articles have been published (DOI: https://doi.org/10.1186/s13045-020-01000-2 and https://doi.org/10.12688/f1000research.8347.1). Therefore, did myeloma has a different observation compared with other cancers?

Round 2

Reviewer 1 Report

The authors accepted all my suggestions and in this form the review is acceptable for publication in IJMS.

Reviewer 2 Report

All questions are satisfied